# Multivariate Exploratory Comparative Analysis of LaLiga Teams: Principal Component Analysis

**DOI:** 10.3390/ijerph18063176

**Published:** 2021-03-19

**Authors:** Claudio A. Casal, José L. Losada, Daniel Barreira, Rubén Maneiro

**Affiliations:** 1Department of Science of Physical Activity and Sport, Catholic University of Valencia “San Vicente Mártir”, 46900 Valencia, Spain; 2Department of Social Psychology and Quantitative Psychology, University of Barcelona, 08001 Barcelona, Spain; jlosada@ub.edu; 3Centre of Research, Education, Innovation and Intervention in Sport (CIFI2D), Faculty of Sport, University of Porto, 4099-002 Porto, Portugal; dbarreira@fade.up.pt; 4Department of Science of Physical Activity and Sport, Pontifical University of Salamanca, 37001 Salamanca, Spain

**Keywords:** performance analysis, elite football, multivariate analysis, principal component analysis, LaLiga

## Abstract

The use of principal component analysis (PCA) provides information about the main characteristics of teams, based on a set of indicators, instead of displaying individualized information for each of these indicators. In this work we have considered reducing an extensive data matrix to improve interpretation, using PCA. Subsequently, with new components and with multiple linear regression, we have carried out a comparative analysis between the best and bottom teams of LaLiga. The sample consisted of the matches corresponding to the 2015/16, 2016/17 and 2017/18 seasons. The results showed that the best teams were characterized and differentiated from bottom teams in the realization of a greater number of successful passes and in the execution of a greater number of dynamic offensive transitions. The bottom teams were characterized by executing more defensive than offensive actions, showing fewer number of goals and a greater ball possession time in the final third of the field. Goals, ball possession time in the final third of the field, number of effective shots and crosses are the main discriminating performance factors of football. This information allows us to increase knowledge about the key performance indicators (KPI) in football.

## 1. Introduction

The identification of performance factors, understood as variables that define some aspect of performance and that help achieve sports success [1], is essential to try to identify the most appropriate behavior patterns that can lead to success [2] and enable the increase and prediction of performance [3,4]. The analysis of the matches will identify those variables related to success [5], and the grouping and combination of these success indicators of different nature will allow the construction of football performance profiles [4,6]. To obtain both the indicators and the performance profiles, the discriminant analysis of the game between teams of different levels is a very useful tool. However, we are facing a sport of a complex and dynamic nature, which makes the identification of these performance profiles a very difficult task [7] because the success of the game can be associated with multiple factors (physical, technical, tactical, …), some of them being unpredictable or uncontrollable, such as arbitration decisions, individual successes or failures of players, match location, type of competition or even chance. 

Football research has turned to a multitude of performance indicators [8], and some studies have tried to identify them through the comparative analysis of successful and unsuccessful teams [9,10,11,12,13,14,15,16,17]. Some of these works show conflicting results. This may be caused, among other things, by the type and size of the sample, the study design, the selection of the variables and the characteristics of the sport itself. It may also be because most studies identify the success of the teams based on the match outcome [9,16,18,19,20,21,22,23]. This discrimination criterion can cause erroneous results because in this sport, in some matches, the team with the best statistical data does not end up getting the victory since in football a single winning play style does not exist. Several teams with different play styles can get similar results. Therefore, it will be necessary to classify the teams, instead of the match outcome, by their position at the end of the season.

To study KPI and performance profiles in football, it would be necessary to perform nomothetic analysis instead of an ideographic one, as the latter would identify the behavior patterns of a unique team and not of the game. It is necessary, therefore, to conduct a longitudinal analysis of all the teams and matches corresponding to one or several regular seasons and classify the teams according to their final position and not based on the match outcome. In this way, the KPI will be more reliable because they will be less mediated by the factors indicated above, and the teams that obtained a higher performance (higher score) at the end of the season can be explained by the fact that they maintained a more effective behavior. Nevertheless, there are few previous studies in this line [11,24,25,26,27,28]. 

Sometimes, to carry out this type of works, especially when indirect observation methodology is used, we find a very extensive data matrix with many related variables. In this case, it would be beneficial to reduce this matrix for a simpler interpretation and eliminate possible redundant information. However, if the reduction is carried out under some subjective criteria, there is a risk of losing relevant information. Therefore, we need some tool that allows us to objectively reduce the dimensions of a data matrix without losing important information. For this, PCA can be an adequate statistical technique since its aims are to simplify, reduce and structure the initial information obtained [29]. Its application to the tactical analysis of football has been demonstrated in various works with satisfactory results. Specifically, Gómez et al. [30] carried out a study with the aim of identifying the independent and interactive effects of the game location and the final result in the statistics related to the football game according to the area of the field in which they occurred in LaLiga, from 2003 to 2004 and 2007 to 2008 seasons. They identified different profiles in the teams related to the match venue and the match outcome. In the work of Moura et al. [31] two main components were identified in the 2006 World Cup and showed that shots, shots on goal and percentage ball possession are some variables that discriminate among winning, drawing and losing teams. Winter and Pfeiffer [23] identified four dimensions in the UEFA Euro 2012 (game speed, transition play after ball recovery, transition play after ball loss and offense efficiency), concluding that the transition play after losing the ball and the offense efficiency seem to be factors connected directly with the match outcome, as those were important values for a successful discrimination. In [32], the specific aim of their paper was to investigate which factors were most crucial for the match outcome in the Serie A, concluding that shot on target is the performance indicator of the game. In the work of Ric et al. [33], a comparative study of the spatial individual and collective organization of the players was carried out between the first and second half of the game. In the work of Fernández-Crehuet et al. [34] an index was built to measure the performance of Spanish Football league teams, during the 2016/2017 season, combining five dimensions: economic, fans-related, historical, team quality and the season’s data. Authors in [35] managed to identify and differentiate various styles of play of the different teams of the Chenesse Soccer Super League during the 2006 season. One style of play denominated possession, other denominated set pieces attack, counterattacking play and, finally, transitional play.

Therefore, we have not found previous works that the PCA have applied to tactically analyze LaLiga teams, during several seasons, and that have determined the level of performance based on the position they occupied in the leaderboard at the end of the season. Nor have they identified and used components to develop a performance model of the teams of different levels. Consequently, we decided to carry out this study to pursue the following aims: the first aim of the present study was to reduce the size of a large database and group it into new categories without losing information, through the PCA. The second aim was to perform a comparative and predictive performance analysis among the best and bottom teams of LaLiga, using the KPI of each group.

## 2. Materials and Methods

### 2.1. Sample 

In order to carry out this study, 1415 records corresponding to the 2015/16, 2016/17 and 2017/18 seasons of LaLiga have been analyzed. These records belong to the best and bottom teams, ranked according to their final score at the end of the regular league (best teams: the best six teams, qualified in the UEFA Champions League and the Europa League; bottom teams: the three lower teams that descended from a category and the next three worst ranked). Data were obtained from the analysis platform InstatScout and analyzed post event. Instatscout (www.instatscout.com) (accessed on 1 April 2019) is a private platform dedicated to assessing the performance of teams in different world leagues. The information cannot be considered either personal or intimate, as the research consisted solely of naturalistic observations in public places, and it was not anticipated that the recordings would be used in a manner that could cause personal harm. According to the Belmont Report [36], the use of public images for research purpose does not require informed consent or the approval of an ethical committee.

### 2.2. Performance Indicators

To carry out the study, 57 performance indicators were used and divided into three groups, according to the available literature [16,19,28,37,38,39]: (1) 5 indicators related to goal scoring, (2) 34 related with the offensive phase, and (3) 18 related to the defensive phase (Table 1). The operative definitions can be consulted in www.instatscout.com (accessed on 19 March 2021).

### 2.3. Data Reliability

To ensure the reliability of the data, five randomly selected matches were coded by the authors of this study and then compared with those provided by InStat. The Kappa (K) values obtained ranged from 0.92 to 0.97. 

### 2.4. Procedure and Statistical Analysis

To analyze the game of both groups, a descriptive, comparative and predictive analysis of the performance of the variable “EFFECTIVENESS” was carried out and defined as
(1)$$\frac{Goals\;Scored+Shots\;on\;target}{Shots}$$

The analysis started checking whether the set of the 57 used indicators correctly differentiated the best and bottom teams. For this, a linear discriminant analysis (LDA) was applied, which is a well-established machine learning technique for categories. Its main advantages are that the model is interpretable and that the prediction is simple.

Secondly, in each group of teams, a Principal Component Analysis (PCA) was carried out to reduce the set of indicators and work with a more manageable size, reducing the multicollinearity problem. This technique allows to transform the original information into a new set of variables, called PC, without losing any information. The first transformed PC captures the greatest amount of information, and each subsequent PC explains a reduced amount of information. For the calculation of the main components the variables were standardized (mean 0 and standard deviation 1). When calculated on standardized variables, the main components are eigenvectors that are taken from the correlation matrix, and as many different main components as available variables can be obtained. The number of significant principal components (PC) was determined by the conventional criteria, but only those that cumulatively accounted for ≥70% of the explained variance were selected [29]. An additional benefit to PCA is that each PC is uncorrelated, so each one captures distinct information within each individual’s data set.

Following this, the orthogonal varimax rotation (Varimax method) performance by Kaiser [40] was performed to determine the contribution of the original categories to the variance explained for each extracted PC as well as improving interpretability. The original categories that demonstrated PC loadings (PC_L_; i.e., eigenvectors of the covariance matrix) that exceeded ±0.70 were considered indicative of a well-defined relationship with the extracted PC [41,42,43]. The loadings can be interpreted as the weight/importance of each variable in each component; therefore, they help to know what type of information each of the components collects. The purpose of the calculation of the loadings is to identify the linear combinations that best represent the variables *X*_1_, …, *X*_p_. Sean (*Z*_1_, *Z*_2_, …, *Z_M_*), where *M* < *p* is linear combinations of the original p variables, that is *Zm = ∑j = 1pϕjmXj*, where *ϕ*1*m*, *ϕ*2*m*, …, *ϕpm*, *ϕ*1*m*, *ϕ*2*m*, …, *ϕpm* are the constants, or loadings, of the main components (for example, *ϕ*_11_ would correspond to the first loading of the first main component).

To finish, an analysis of variance (ANOVA) was carried out to check the differences between the different main components. By means of multiple linear regression, a prediction model was also constructed for each group, formed by the explained variable “EFFECTIVENESS”, and the main components found in each group.
(2)$$Y_{t}=\;\beta_{0}+\beta_{1}X_{1}+\beta_{2}X_{2}+\ldots +\;\beta_{p}X_{p}+\;\varepsilon \;+\;Y_{t}\;explained\;variable;X_{p}\;principals\;componts\;of\;each\;variable$$

This model will allow us to identify the set of variables that have greater influence on the performance of each group and check if there are differences between the styles of play between both groups of teams.

To carry out the statistical analysis, the R program (v.3.4.1) was used, using the MASS [44], stats and citatiom (“factomineR and factoextra”) libraries.

## 3. Results

The results obtained from the linear discriminant analysis (LDA) were 85.63% well classified and 14.36% poorly classified, which reveals that the indicators used correctly classified the teams as best and bottom.

### 3.1. Best Teams PCA Results

Figure 1 shows the screeplot, the eigenvectors and the accumulation of the variance explained by each PC. The eight eigenvectors produced by the PCA explained 70.1% of the total variance, and its eigenvalues were higher than 1 (see Table 2).

The component loadings after rotation is illustrated in Table 3. Applying the Varimax rotation method maximizes the variance of the matrix of charges so that the values are more interpretable. The “rotated” matrix provides the loadings of the main components, and each column contains the vector of loadings for each main component.

The information in this table can be summarized as shown in Figure 2. The dashed horizontal line indicates the average contribution value of the different categories in each component. A category with a value greater than this limit is considered important for that component. The dominant category (or categories) in each component was used to subjectively characterize new variables: (a) Passes; (b) Challenges; (c) Attacks effectiveness; (d) Shots; (e) Dribbles; (f) Tackles; (g) Offensive transitions and (h) Possession.

### 3.2. Bottom Teams PCA Results

In bottom teams a screeplot was generated to represent the eigenvectors ordered from highest to lowest (Figure 3).

Taking into account the information of the accumulated variance and that of the eigenvalues, the first nine components that explain 70% of the variance were used (Table 4). Table 5 shows the bottom teams’ component loadings after rotation.

Figure 4 shows the contribution of the different categories in each component and the new subjectively defined category: (a) Passes; (b) Challenges; (c) Shots; (d) Attacks effectiveness; (e) 1 vs. 1; (f) Dribbles; (g) Fouls; (h) Possession balls and (i) Offensive transitions.

### 3.3. Best Teams ANOVA and Linear Regression Model

An analysis of variance indicated that there were significant differences in the eight main components (Table 6).

It was verified that the residues were distributed randomly around 0. The Shapiro–Wilk test was performed to check the normality of the main component residuals. A *p*-value = 0.337 was obtained. Therefore, they follow normality. For the homoscedasticity of the residues the test of Breusch–Pagan was used with a value of BP = 1.8871, df = 2, and a *p*-value = 0.389, which concludes that there is no evidence of lack of homoscedasticity.

Autocorrelation was studied by means of the DW statistic that showed a result of 1879, with a *p*-value = 0.959. Therefore, there is no evidence of autocorrelation. The eight main components were entered into a linear regression model (Table 7) to predict the explained variable “EFFECTIVENESS”. It can be seen how all the PCs provided different information to the model, with PC3 being the one with the greatest weight (0.76260) and PC1 with the lowest weight (0.03851).

The model conformed to the observed data (adjusted R-squared: 0.6813), and its main components were significant, *p*-value: <0.000. The multiple linear regression model would be
ξ (EFFECTIVENESS) = 7.777 + 0.038 CP1 − 0.23416 CP2 + 0.76260 CP3 + 0.36481 CP4 + 0.22498 CP5 − 0.40160 CP6 + 0.40160 CP7 + 0.334 CP8(3)

### 3.4. Bottom Teams ANOVA and Linear Regression Model

In Table 8 we can see how significant differences have been identified among the nine main components of the bottom teams.

Residuals must be distributed randomly around 0. The Shapiro–Wilk test affirmed the normality of the residuals (*p*-value = 0.262). The Breusch–Pagan test values were BP = 1.8871, df = 2, and a *p*-value = 0.389, from which it follows that there was no evidence of a lack of homoscedasticity. The statistic values D-W = 1989, *p*-value 0.972, showed no evidence of autocorrelation.

Once the conditions in the residuals were checked, the linear regression model in unsuccessful teams was established (Table 9).

The model conformed to the observed data (adjusted R-squared: 0.6813), and its main components were significant, *p*-value: <0.000. The multiple linear regression model would be
ξ (EFFECTIVENESS) = 6.727 + 0.175 CP1 + 0.070 CP2 + 0.308 CP3 − 0.842 CP4 − 0.18340 CP5 + 0.559 CP6 − 0.54061 CP7 + 0.513 CP8(4)

## 4. Discussion

To identify the indicators that influence football performance we perform a comparative analysis between teams of different levels of success, but sometimes we find a set of data with many related categories; therefore, the application of techniques that reduce the quantity of data could be useful. In this work we have considered reducing the dimensions of a data matrix without the loss of relevant information, using PCA. Subsequently we have used these PCs to try to identify the difference in performance between the best and bottom teams of LaLiga.

The PCA data mining technique allowed reducing the dimensions of a broad set of original categories without losing information, creating new categories for both groups. Specifically, we managed to reduce the original data matrix, composed of 57 categories in less than 10 new categories and with an explanation of the variance of ≥70%, enabling the grouping of information and the simplification of the analysis. For the best teams group, PCA created 8 PCs that explained 70.1% of the variance (Table 1): Passes (0.27%); Challenges (0.15%); Attack effectiveness (0.08%); Shots (0.07%); Dribbles (0.04%); Tackles (0.04%); Offensive transitions (0.03%) and Possession (0.03%). For the bottom teams group, 9 PCs were created that explained 70% of the variance (Table 3): Passes (0.23); Challenges (0.14%); Shots (0.08%); Attacks effectiveness (0.06%); 1vs1 (0.05%); Dribbles (0.04%); Fouls opponent (0.04%); Possession (0.03%) and Offensive transitions (0.03%).

In both groups, the Passes PC is denominated this way because most of the categories that constitute it refer to the number of passes and the time of possession. Challenges PC received this name because it included all types of challenges. The Attack effectiveness PC collected categories of the offensive phase, especially related to goals, shots and the effectiveness of shots. The Shots PC mainly included categories related to goals, shots, possession and passes. The Dribbles PC was mainly constituted by categories referring to dribbling, tackles and challenges. The Tackles PC was related to dribbling, challenges, tackles and lost balls. The Offensive transitions PC received this name for being related to recoveries, interceptions and counterattacks. The Possession PC, in the group of successful teams, is the one that showed a worse definition since it is made up of categories with less relation between them. In the bottom teams the 1vs1 PC included all dribbles and tackles. Fouls opponent PC is constituted by varied categories, being the heaviest ones the fouls opponent and, finally, Possession PC is also formed by different categories, the time of possession being the most important.

Therefore, the PCA was shown, as in some previous works [23,31,33,34,35,45,46,47], as a good statistical technique, when we intend to reduce large data sets that have many interrelated variables, allowing us not only to speak of individual performance indicators, but of a set of related indicators.

If we use the PCs to compare the game of both groups, the first difference we observe is that, to explain the same percentage of variance, for the best teams group we need eight PCs, and for the bottom teams we need nine PCs. In both groups, both the category constituted from PC and called Passes, as well as Challenges, were those that allowed explaining the highest percentage of the variance. The Passes category had a slightly greater weight (27%) in the best teams group than in the bottom teams (23%) (Table 2 and Table 4). On the other hand, Challenges showed a similar weight in both groups (15% and 14%). However, the loadings of each PC were not exactly the same for each group (Table 3 and Table 5). Thus, for Passes PC in the best teams group, the most important categories were passes, passes accurate, passes accurate left and passes accurate right. For the bottom teams, the highest weight categories were possession, passes, passes forward, passes left, passes right and passes forward accurate. Therefore, we can indicate that successful teams are characterized more by the efficiency of the passes than by the number of passes executed. That is, they have a greater number of successful passes than lower level teams. These results coincide with some previous works [22], but they analyzed 2014 Brazil FIFA World Cup and used a logistic regression. For Challenges PC we have also found some differences. It can be seen how, for the best teams, the attack challenges had greater weight; however, the defensive challenges were the ones most relevant for bottom teams. This circumstance can be explained because the bottom teams are characterized by staying longer in the defensive phase, executing many more defensive than offensive actions. Previous work also coincides in indicating that the successful teams show higher averages of offensive variables, and unsuccessful teams show higher averages of defensive variables [48].

Another difference that we can see in terms of PC formation is that in the best teams the PC called Tackles is formed, consisting mainly of the categories dribbling, challenges, tackles and lost balls. In the bottom teams the 1vs1 PC and fouls were constituted but did not appear in the other group. In spite of these differences we can appreciate that both the components constituted for both groups, as well as the categories and the weight of these in each component, were very similar. This circumstance leads us to think that in high level football the differences between the teams are minimal, and their success or failure may be explained by the individual performance of their players.

The results of the linear regression model (Table 5 and Table 7) allow us to identify which PCs have the greatest influence on the performance of both groups of teams. For this, a prediction model of the category “EFFECTIVENESS” was built, both for the best and for the bottom teams. The linear regression model of the best teams group, ordering the PCs from highest to lowest weight, was constituted as follows: Attack effectiveness (0.76260); Offensive transitions (0.40160); Shots (0.36481); Possession (0.33451); Dribbles (0.22498); Passes (0.03851); Challenges (−0.23416) and Tackles (−0.40160). In the bottom teams the order was as follows: Dribbles (0.55955); Possession (0.51367); Shots (0.30873); Passes (0.17582); Challenges (0.07051); 1vs1 (−0.18340); Fouls (−0.54061) and Attack effectiveness (−0.84295). We can see how in best teams, the PC that offered a greater influence on the prediction of this category was Attack effectiveness. The number of goals, a greater ball possession time in the final third of the field, a greater number of effective shots and crosses allow to increase the performance in best teams. This information is essential for technicians since, if they manage to improve the performance of their teams in these elements of the game, they will increase their offensive performance. The information provided by the number of goals is trivial since it is obvious that scoring more goals implies increasing offensive performance, but the other indicators referring to ball possession zone, effective shots and crosses do offer transcendent information. These results are corroborated by the works of [9,10] who indicated that successful teams have longer-term possessions in the middle of the offensive field than the defensive one. The works [19,22,49,50] indicate that successful teams show greater effectiveness in shooting, also ratify in their work that making a greater number of crosses increases the chances of winning the matches. In contrast to the cited studies, in our work we have obtained similar results using a different method, specifically through a data mining technique. Winter and Pfeiffer [23] also reached the same conclusion in their work, indicating that there is a relationship between offense efficiency and success, but they analyzed UEFA Euro 2012 and considered success as the match outcome.

Following the results of the linear regression model, we can indicate how the main differences in the prediction of performance of both groups occur in PCs offensive transitions, tackles, challenges, dribbles, fouls opponent and 1vs1. Offensive transitions play a more important role in the best teams than in the bottom teams. Thus, in the best teams, performing a greater number of recoveries, interceptions and counterattacks, that is, dynamic offensive transitions through counterattacks, would increase their performance in the game. This circumstance was also pointed out by Tenga et al. [40]. These authors analyzed the Norwegian league, and by means of a multiple linear regression, they obtained that the proportion of goals scored during counterattacks (52%) was higher than during elaborate attacks (48%). Therefore, the offensive game seems to be more efficient against a disorderly defense. This information is very important for the coaches, who should focus their training on these game situations, both in attack and defense, to try to improve their performance in both phases of the game.

In the best teams the Tackles and Challenges PCs negatively influenced the offensive performance. This may be due to the fact that these are more typical behaviors of unsuccessful teams, as indicated above [48].

In bottom teams it was appreciated how increasing the number of successful dribblings would increase performance. This result coincides with that of the work of Harrop and Nevill [21] who found that the number of dribbles is correlated with performance. The PC Fouls opponent also showed a strong negative influence on the performance of bottom teams and that these teams showed fewer effective attacks than the best teams. 

We have achieved the aims set and the sample used, as these are the matches of three competitive seasons, allowing us to generalize the results. The main contribution and novelty of this work is that we have carried out a longitudinal tactical analysis of LaLiga teams, using the combination of factor analysis and linear regression. However, we believe that the differences found in the constitution of the different PCs have not been as satisfactory as we would have liked. We believe that this may be due to the design used, in our case we have found the PCs for each group of teams separately and, subsequently, we have tried to build a probabilistic model with the detected PCs. In future works we should propose a design in which we find the main components for both groups and then build a separate model for each group. In addition, since the goals scored and received did not have a significant contribution to the main components, in the future it could be considered to eliminate these variables from the analysis because this approach may be biasing the same.

The results of this work offer information to the technicians, about what are the KPIs in football and the game pattern of the best teams, being able to compare the latter with that of their own teams, and thus, to be able to make the appropriate modifications, to increase performance.

## 5. Conclusions

The realization of this work has allowed us, with the use of the PCA, to reduce a dimension of data without losing relevant information. We have been able to identify the KPI of the best and bottom teams, and we have identified the main differences between both groups. 

Best teams are characterized and differentiated from bottom teams in the realization of a greater number of successful passes and in the execution of a greater number of dynamic offensive transitions.

Bottom teams are characterized by executing more defensive than offensive actions.

A greater ball possession time in the final third of the field, and a greater number of effective shots and crosses, are the main performance factors that influence the offensive success of football.

## Figures and Tables

**Figure 1 ijerph-18-03176-f001:**
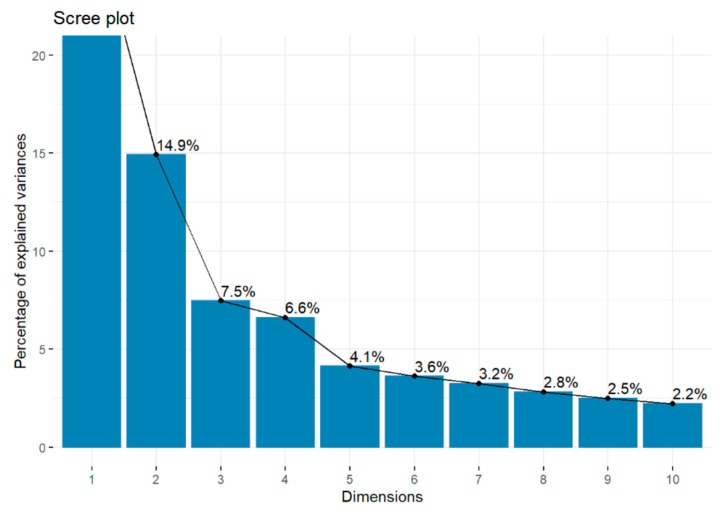
Screeplot of best teams.

**Figure 2 ijerph-18-03176-f002:**
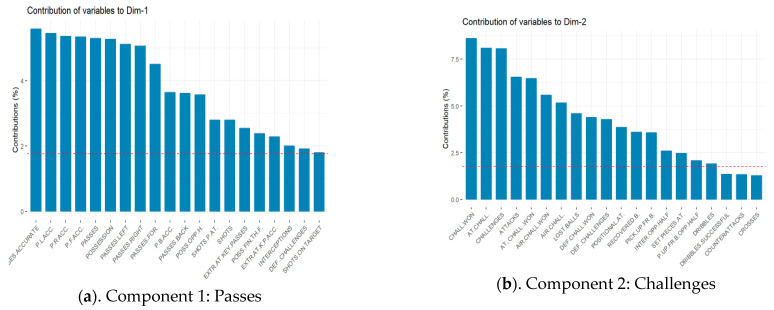
Components and new subjective variables of best teams. (**a**): Passes; (**b**): Challenges; (**c**): Attack effectiveness; (**d**): Shots; (**e**): Dribbles; (**f**): Tackles; (**g**): Offensive transitions; (**h**): Possession.

**Figure 3 ijerph-18-03176-f003:**
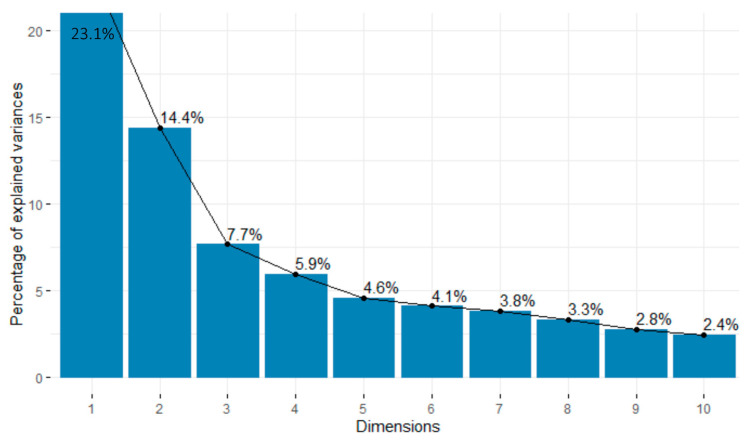
Screeplot for bottom teams.

**Figure 4 ijerph-18-03176-f004:**
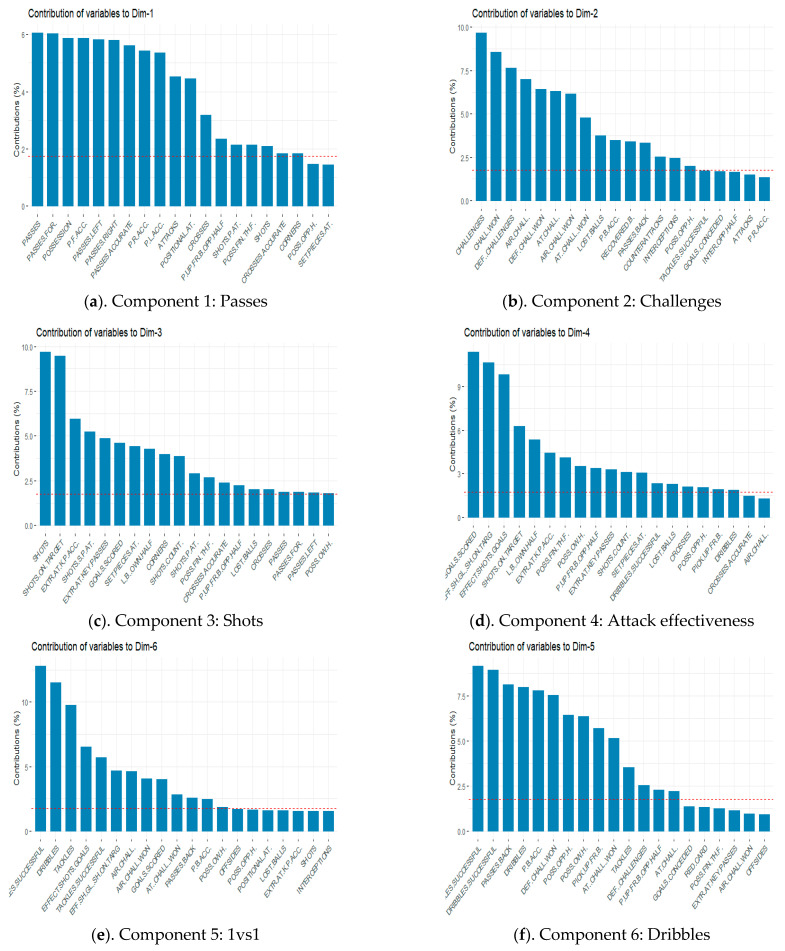
Components and new subjective variables of bottom teams. (**a**): Passes; (**b**): Challenges; (**c**): Shots; (**d**): Attacks effectiveness; (**e**): 1 vs 1; (**f**): Dribbles; (**g**): Fouls; (**h**): Possession balls; (**i**): Offensive transitions.

**Table 1 ijerph-18-03176-t001:** Selected performance indicators.

Outcome of Attack	Offence	Defence
Goals scored, Shots, Shots on target, Effectiveness shots goals, Effectiveness shots goals and shots on target.	Attacks, Positional attacks, Attacks with shots-positional, Counterattacks, Counterattacks with shots, Set pieces attack, Attacks with shot set pieces attacks, Ball possession, Ball possession in own half, Ball possession in opponent half, Ball possession time in the final third of the field, Attacking challenges, Attacking challenges won, Corners, Crosses, Crosses accurate, Dribbles, Dribbles successful, Fouls opponent, Lost balls, Lost balls in own half, Offsides, Passes, Passes accurate, Extra attacking and key passes, Extra attacking and key passes accurate, Passes forward, Passes forward accurate, Passes back, Passes back accurate, Passes to the left, Passes to the left accurate, Passes to the right, Passes to the right accurate.	Challenges, Challenges won, Air challenges, Air challenges won, Defensive challenges, Defensive challenges won, Fouls, Goals conceded, Interceptions, Interceptions in opposition half, Picking up free balls, Picking up free balls in opposition half, Recovered balls, Recovered balls in opposition half, Tackles, Tackles successful, Red cards, Yellow cards.

**Table 2 ijerph-18-03176-t002:** Components and total variance explained.

Parameters	PC1	PC2	PC3	PC4	PC5	PC6	PC7	PC8
Standard deviation	3.94	2.91	2.06	1.93	1.53	1.43	1.35	1.26
Proportion of Variance	0.27	0.14	0.07	0.06	0.04	0.03	0.03	0.02
Cumulative Proportion	0.27	0.42	0.49	0.56	0.60	0.64	0.67	0.70

**Table 3 ijerph-18-03176-t003:** PCA factorial matrix of best teams.

Categories	PC1	PC2	PC3	PC4	PC5	PC6	PC7	PC8
GOALS SCORED	−0.08	0.09	**−0.26**	**0.23**	0.08	**−0.14**	0.10	−0.09
GOALS CONCEDED	0.02	−0.06	0.10	0.02	**−0.14**	0.01	0.01	**0.16**
YELOW CARD	0.07	−0.01	0.09	−0.06	−0.07	0.09	0.00	−0.01
RED CARD	0.05	0.03	0.05	−0.01	**−0.14**	0.04	−0.08	0.11
CORNERS	0.10	−0.09	**0.20**	**0.19**	0.07	0.11	−0.02	**−0.15**
SHOTS SET PIECES ATTACK	−0.10	−0.07	**0.13**	**0.19**	−0.07	−0.11	0.02	**0.15**
OFFSIDES	−0.02	−0.02	−0.06	0.11	−0.03	−0.12	**0.14**	0.00
FOULS	0.10	−0.00	0.09	0.00	0.04	0.12	0.05	0.00
FOULS OPPONENT	−0.02	−0.09	−0.01	−0.05	**−0.31**	0.07	**0.23**	−0.01
POSSESSION	**−0.22**	−0.06	−0.03	−0.10	0.01	−0.01	−0.03	−0.01
POSSESSION OWN HALF	−0.09	0.02	**−0.21**	**−0.14**	−0.03	**−0.19**	−0.05	**0.43**
POSSESSION OPPONENT HALF	**−0.18**	0.03	**0.13**	−0.05	0.05	0.12	0.08	0.05
POSSESSION FINAL THIRD FIELD	**−0.15**	−0.04	**0.22**	0.06	0.05	**0.15**	0.05	**−0.32**
SHOTS ON TARGET	**−0.13**	0.04	−0.12	**0.31**	0.04	−0.07	0.06	−0.04
EFF. SHOTS GOALS	−0.01	−0.09	**−0.30**	0.11	0.09	**−0.18**	**0.15**	−0.11
EFFECTIVENESS	−0.01	0.09	**−0.30**	**0.14**	0.08	**−0.15**	**0.14**	−0.13
ATTACKS	−0.10	**−0.25**	0.08	0.04	−0.03	**−0.15**	−0.02	0.05
POSITIONAL ATTACK	**−0.13**	**−0.19**	0.06	−0.11	−0.03	**−0.15**	0.09	−0.00
SHOTS PIECES ATTACK	**−0.16**	0.00	0.04	**0.16**	0.02	−0.01	0.06	−0.01
COUNTERATTACKS	0.06	−0.11	−0.07	**0.21**	0.10	0.03	**−0.33**	0.00
SHOTS COUNTERATTACK	−0.03	0.03	−0.03	**0.29**	**0.13**	0.06	**−0.23**	−0.06
SET PIECES ATTACK	−0.07	**−0.15**	**0.20**	**0.18**	**−0.13**	−0.11	0.07	**0.17**
PASSES	**−0.23**	−0.06	−0.06	−0.13	0.05	0.01	−0.04	−0.06
PASSES ACCURATE	**−0.23**	−0.02	−0.06	−0.11	0.04	0.04	−0.02	−0.06
EXTRA AT. KEY PASSES	**−0.15**	−0.01	−0.11	**0.20**	−0.05	−0.07	0.02	0.06
EXTRA AT. KEY PASSES ACCURATE	**−0.15**	−0.01	−0.11	**0.20**	−0.05	−0.07	0.02	0.06
CROSSES	−0.10	−0.11	**0.27**	0.06	0.12	−0.06	−0.00	−0.01
CROSSES ACCURATE	−0.08	−0.08	**0.22**	0.11	**0.13**	−0.08	−0.02	−0.04
CHALLENGES	0.10	**−0.28**	−0.08	−0.00	0.00	0.08	0.07	−0.02
CHALLENGES WON	0.05	**−0.29**	−0.12	0.03	−0.02	0.11	**0.16**	−0.06
DEFFENCE CHALLENGES	**0.13**	**−0.20**	−0.07	−0.01	**0.17**	0.10	**0.15**	−0.00
DEFFENCE CHALLENGES WON	0.09	**−0.20**	−0.06	−0.02	**0.23**	0.05	**0.28**	−0.00
AT.CHALLENGES	0.04	**−0.28**	−0.07	0.00	**−0.18**	0.04	−0.03	−0.03
AIR CHALLENGES	0.11	**−0.22**	0.03	−0.07	0.03	**−0.24**	0.08	**−0.15**
AIR CHALLENGES WON	0.08	**−0.23**	0.02	−0.05	0.03	**−0.23**	**0.14**	**−0.15**
DRIBBLES	−0.09	**−0.13**	**−0.18**	−0.05	**−0.32**	**0.25**	−0.06	0.02
DRIBBLES SUCCESSFUL	−0.10	−0.11	**−0.18**	0.08	**−0.33**	**0.25**	−0.04	0.00
TACKLES	0.10	−0.06	−0.09	0.09	**0.24**	**0.36**	0.09	**0.24**
TACKLES SUCCESSFUL	0.07	−0.09	−0.10	0.07	**0.31**	**0.31**	**0.20**	**0.22**
INTERCEPTIONS	0.14	−0.07	−0.09	−0.05	0.03	−0.05	**−0.31**	−0.10
INTERCEPTIONS OPPOSITION HALF	−0.01	**−0.16**	0.01	0.02	0.07	−0.00	**−0.29**	−0.01
PICKING UP FREE BALLS	0.01	**−0.18**	0.05	−0.06	**0.13**	−0.12	0.08	0.13
PICKING UP FREE BALL OPPOSITION HALF	−0.12	**−0.14**	0.15	0.02	0.11	−0.03	0.04	**0.17**
LOST BALLS	0.07	**−0.21**	−0.10	−0.05	0.00	**−0.26**	**−0.14**	**0.21**
LOST BALLS OWN HALF	0.10	−0.04	**−0.21**	−0.09	−0.09	−0.06	**−0.21**	−0.09
RECOVERED BALL	0.04	**−0.18**	−0.09	0.03	**0.17**	0.01	**−0.30**	−0.01
RECOVERED BALL OPPOSITION HALF	−0.06	−0.08	0.04	0.12	**0.18**	0.10	−0.26	**0.17**
PASSES FORWARD	**−0.21**	−0.10	−0.06	**−0.14**	−0.08	−0.00	−0.00	−0.06
PASSES BACK	**−0.19**	0.07	−0.04	−0.12	0.05	0.02	0.08	**0.24**
PASSES LEFT	**−0.22**	−0.05	−0.06	**−0.14**	0.07	0.02	−0.03	−0.05
PASSES RIGHT	**−0.22**	−0.05	−0.07	**−0.14**	0.06	0.01	−0.02	−0.07
PASSES FORWARD ACCURATE	**−0.23**	−0.05	−0.05	−0.11	0.05	0.04	−0.04	−0.09
PASSES BACK ACCURATE	**−0.19**	0.08	−0.05	−0.13	0.05	0.02	0.08	**0.23**
PASSES LEFT ACCURATE	**−0.23**	−0.02	−0.06	−0.12	0.06	0.05	−0.02	−0.05
PASSES RIGHT ACCURATE	**−0.23**	−0.02	−0.06	−0.13	0.05	0.04	−0.01	−0.06

Component *loadings* ≥ median contribution appear in bold. EFF: Effectiveness. SH: Shot. AT: Attack.

**Table 4 ijerph-18-03176-t004:** Bottom teams’ principal components and total variance explained.

Parameters	PC1	PC2	PC3	PC4	PC5	PC6	PC7	PC8	PC9
Standard deviation	3.65	2.86	2.08	1.84	1.61	1.52	1.47	1.37	1.25
Proportion of Variance	0.23	0.14	0.07	0.05	0.04	0.04	0.03	0.03	0.03
Cumulative Proportion	0.23	0.37	0.45	0.51	0.56	0.60	0.63	0.67	0.70

**Table 5 ijerph-18-03176-t005:** PCA factorial matrix of bottom teams.

Categories	PC1	PC2	PC3	PC4	PC5	PC6	PC7	PC8	PC9
GOALS SCORED	0.01	−0.01	**0.21**	**−0.33**	−0.05	**0.19**	**−0.18**	**0.13**	−0.06
GOALS CONCEDED	0.02	0.12	−0.01	−0.03	0.11	−0.09	0.04	−0.06	−0.19
YELOW CARD	0.03	−0.06	0.02	0.03	0.04	−0.00	−0.07	0.00	0.12
RED CARD	0.04	0.00	−0.00	0.04	0.11	0.02	−0.02	−0.03	**−0.25**
CORNERS	**−0.13**	0.02	**0.19**	0.08	0.03	0.00	0.02	**−0.25**	−0.13
OFFSIDES	−0.01	−0.05	−0.06	−0.09	−0.09	0.12	0.06	0.08	0.04
FOULS	0.06	−0.09	0.04	0.11	−0.07	−0.05	−0.09	0.08	0.10
FOULS OPPONENT	−0.07	−0.06	−0.02	0.06	0.08	−0.00	**−0.30**	0.15	0.04
POSSESSION	**−0.24**	0.06	−0.07	−0.05	0.01	0.05	0.03	0.02	0.02
POSSESSION OWN HALF	−0.04	0.04	−0.13	**−0.18**	**−0.25**	**−0.13**	**−0.21**	**−0.37**	−0.01
POSSESSION OPPONENT HALF	−0.12	**0.14**	0.07	**0.14**	**−0.25**	−0.12	**−0.16**	0.02	**0.24**
POSSESSION FINAL THIRD FIELD	**−0.14**	0.04	**0.16**	**0.20**	0.11	0.08	0.12	**0.31**	0.10
SHOTS ON TARGET	−0.05	−0.00	**0.30**	**−0.25**	−0.01	0.01	−0.03	−0.08	−0.08
EFF. SHOTS GOALS	0.06	−0.02	0.10	**−0.31**	−0.07	**0.25**	**−0.23**	**0.16**	−0.01
EFFECTIVENESS	0.06	−0.02	0.11	**−0.32**	−0.07	**0.21**	**−0.20**	**0.19**	0.01
ATTACKS	**−0.21**	−0.12	0.06	0.05	0.01	0.09	−0.03	−0.11	0.00
POSITIONAL ATTACK	**−0.21**	−0.05	−0.04	0.05	−0.01	0.12	−0.11	−0.01	−0.06
SHOTS PIECES AT.	**−0.14**	0.05	0.17	−0.05	0.01	−0.09	0.06	0.01	**−0.15**
COUNTERATTACKS	−0.00	**−0.15**	0.11	−0.11	0.00	−0.04	**0.24**	−0.11	**0.23**
SHOTS COUNTERATTACK	0.00	−0.03	**0.19**	**−0.17**	−0.01	−0.08	**0.23**	−0.02	0.07
SET PIECES ATTACK	−0.12	−0.06	**0.21**	**0.17**	0.06	0.00	−0.11	**−0.20**	−0.11
PASSES	**−0.24**	0.07	−0.13	−0.07	0.02	0.08	0.07	0.04	0.00
PASSES ACCURATE	**−0.23**	0.11	−0.12	−0.06	0.02	0.04	0.05	0.08	−0.00
EXTRA AT. KEY PASSES	−0.10	0.01	**0.22**	**−0.18**	−0.10	−0.10	0.02	−0.07	0.03
EXTRA AT. KEY PASSES ACCURATE	−0.08	0.01	**0.24**	**−0.21**	−0.09	−0.12	0.02	−0.02	0.02
CROSSES	**−0.17**	0.03	**0.14**	**0.14**	0.00	0.07	0.08	−0.01	−0.12
CROSSES ACCURATE	**−0.13**	0.01	**0.15**	0.12	−0.00	0.05	−0.00	0.03	**−0.20**
CHALLENGES	−0.07	**−0.31**	−0.07	0.01	0.00	−0.07	−0.04	0.07	−0.04
CHALLENGES WON	−0.09	**−0.29**	−0.03	0.00	−0.02	−0.10	−0.08	0.11	−0.03
DEFFENCE CHALLENGES	−0.02	**−0.27**	−0.10	−0.00	**−0.16**	−0.02	0.10	0.12	−0.12
DEFFENCE CHALLENGES WON	−0.04	**−0.25**	−0.06	0.04	**−0.27**	−0.00	0.03	**0.13**	−0.10
AT.CHALLENGES	−0.10	**−0.25**	−0.02	0.03	**0.14**	−0.10	**−0.18**	−0.00	0.04
AIR CHALLENGES	−0.06	**−0.26**	−0.02	0.11	−0.08	**0.21**	−0.12	−0.05	−0.00
AIR CHALLENGES WON	−0.07	**−0.24**	−0.00	0.09	−0.09	**0.20**	**−0.14**	−0.00	−0.00
DRIBBLES	−0.10	−0.09	−0.05	**−0.13**	**0.28**	**−0.33**	−0.11	0.03	0.10
DRIBBLES SUCCESSFUL	−0.10	−0.08	−0.03	**−0.15**	**0.29**	**−0.35**	**−0.15**	0.03	0.08
TACKLES	0.05	−0.11	−0.07	−0.11	**−0.18**	**−0.31**	**0.21**	**0.16**	**−0.21**
TACKLES SUCCESSFUL	0.00	**−0.13**	−0.06	−0.03	**−0.30**	**−0.23**	**0.18**	**0.19**	**−0.19**
INTERCEPTIONS	0.07	**−0.15**	−0.05	−0.09	0.07	0.12	0.10	**−0.18**	**0.33**
INTERCEPTIONS OPPOSITION HALF	−0.07	−0.12	0.07	0.02	0.02	0.05	0.09	**−0.15**	**0.25**
PICKING UP FREE BALLS	−0.10	−0.11	0.02	**0.13**	**−0.23**	0.02	−0.03	−0.05	−0.02
PICKING UP FREE BALL OPPOSITION HALF	**−0.15**	−0.03	**0.14**	**0.18**	**−0.15**	−0.05	−0.03	−0.04	−0.01
LOST BALLS	−0.05	**−0.19**	**−0.14**	**−0.15**	0.08	0.12	0.06	**−0.34**	**−0.21**
LOST BALLS OWN HALF	0.03	−0.09	**0.20**	**−0.23**	0.05	0.06	0.04	**−0.28**	**−0.24**
RECOVERED BALL	−0.08	**−0.18**	0.01	−0.08	−0.07	0.05	**0.24**	−0.04	**0.28**
RECOVERED BALL OPPOSITION HALF	−0.09	−0.06	0.13	−0.00	−0.07	−0.01	**0.20**	−0.05	**0.19**
PASSES FORWARD	**−0.24**	0.01	−0.13	−0.06	0.02	0.11	0.09	0.02	0.01
PASSES BACK	−0.11	**0.18**	−0.08	−0.03	**−0.28**	**−0.16**	**−0.20**	−0.12	0.11
PASSES LEFT	**−0.24**	0.06	−0.13	−0.07	0.02	0.07	0.06	0.04	0.00
PASSES RIGHT	**−0.24**	0.07	−0.13	−0.06	0.01	0.08	0.07	0.04	0.00
PASSES FORWARD ACCURATE	**−0.24**	0.08	−0.11	−0.06	0.04	0.05	0.06	0.07	−0.00
PASSES BACK ACCURATE	−0.11	**0.18**	−0.08	−0.03	**−0.27**	**−0.15**	**−0.20**	−0.11	0.10
PASSES LEFT ACCURATE	**−0.23**	0.11	−0.13	−0.07	0.04	0.03	0.04	0.08	0.00
PASSES RIGHT ACCURATE	**−0.23**	0.11	−0.11	0.07	0.01	0.04	0.05	0.07	−0.01

Component *loadings* ≥ median contribution appear in bold. EFF: Effectiveness. SH: Shot. AT: Attack.

**Table 6 ijerph-18-03176-t006:** ANOVA best teams.

Model	Df Sum	Sq Mean	Sq F	Value	Pr (>*F*)
Pca1	8	2966	370.8	183.5	<0.000 ***
Residuals		675	1364	2.0	

Signif. codes: *** 0.001.

**Table 7 ijerph-18-03176-t007:** Best teams linear regression model.

Components	Estimate Std	Error	*t* Value	Pr (>|t|)
(Intercept)	7.77778	0.05435	143.099	<0.000 ***
pca1PC1	0.03851	0.01377	2.796	0.00532 **
pca1PC2	−0.23416	0.01866	−12.551	<0.000 ***
pca1PC3	0.76260	0.02638	28.908	<0.000 ***
pca1PC4	0.36481	0.02806	13.002	<0.000 ***
pca1PC5	0.22498	0.03547	6.342	<0.000 ***
pca1PC6	−0.40160	0.03793	−10.588	<0.000 ***
pca1PC7	0.40160	0.03793	10.588	<0.000 ***
pca1PC8	0.33451	0.04307	7.767	<0.000 ***

Residual standard error: 1.422 on 675 degrees of freedom; Multiple R-squared: 0.685; Adjusted R-squared: 0.6813; F-statistic: 183.5 on 8 and 675 DF, *p*-value: < 0.000. Signif. codes: *** < 0.000, ** < 0.001.

**Table 8 ijerph-18-03176-t008:** ANOVA bottom teams.

Model	Sum of Squares	Df	Mean Square	F	Sig.
Regression	86,366	9	9596	357,605	0.000 ^b^
Residual	37,434	1395	0.027		
Total	123,800	1404			

Dependent Variable: EFFECTIVENESS; ^b^. Predictors: (Constant), BottomPC1, BottomPC2, BottomPC3, BottomPC4, BottomPC5, BottomPC6, BottomPC7, BottomPC8, BottomPC9.

**Table 9 ijerph-18-03176-t009:** Bottom teams linear regression model.

Components	Estimate Std	Error	*t* Value	Pr(>|t|)
(Intercept)	6.72715	0.04504	149.352	<0.000 ***
pca1PC1	0.17582	0.01232	14.273	<0.000 ***
pca1PC2	0.07051	0.01575	4.478	<0.000 ***
pca1PC3	0.30873	0.02157	14.312	<0.000 ***
pca1PC4	−0.84295	0.02448	−34.433	<0.000 ***
pca1PC5	−0.18340	0.02786	−6.582	<0.000 ***
pca1PC6	0.55955	0.02951	18.964	<0.000 ***
pca1PC7	−0.54061	0.03054	−17.704	<0.000 ***
pca1PC8	0.51367	0.03279	15.665	<0.000 ***
pca1PC9	0.05140	0.03587	1.433	0.152

Signif. codes: *** <0.000.

## Data Availability

Data was obtained from InStatscout and are available from www.instatscout.com (accessed on 19 March 2021).

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
