# Peer review of "Multivariate Exploratory Comparative Analysis of LaLiga Teams: Principal Component Analysis"

_ijerph, 2021, doi:10.3390/ijerph18063176_

Round 1
Reviewer 1 Report
Dear authors, first of all I would like to congratulate you for the work done, it is worth noting the number of records analyzed (1472). However, I consider that the study is not enough to be published in IJERPH for different reasons:
- The results are not original, since they have been previously reported, many of them in studies cited in the manuscript itself (22, 42…).
- In the discussion section, the authors describe or report the results obtained, again similar to those reported by previous referenced studies (it is correct, they justify them), but they do not debate or discuss the reasons for those results or no indications are given of how they could be applied by coaches. Example paragraphs:243-253,254-267, 293-300…
- Review table 1: goals conceded.
- Line 276-277: The authors could justify that 4 points is a “slightly greater weight”
- Line 286: References.
- I recommend a more careful writing of the conclusions, since the authors have reported results more than conclusions.
Author Response
Dear reviewer, thank you very much for your comments and suggestions, we indicate the changes we have made based on the review made.
Reviewer
The results are not original, since they have been previously reported, many of them in studies cited in the manuscript itself (22, 42…).
Authors
We regret to indicate that we do not completely agree with your statement, because, first of all, we consider that a research work does not stop being interesting for the scientific community or for society due to the fact that it does not report results different from those of previous works, in this case, it would serve to corroborate these results.
On the other hand, we do not exactly coincide with your statement, since the first objective that we set in this work for ourselves is to check if we can reduce a very large data matrix without losing important information, by PCA. The fact that some of our results coincide with some previous work only confirms the achievement of this objective.
On the other hand, the method used in this work differs considerably from the previous ones. In this work, a longitudinal and nomoteticific analysis is carried out, the sample is different from that of the cited works, we apply a different statistical technique (PCA), and we classify the teams by the final position in the classification table.
The number of main components and the variables that compose them are not the same as in previous works. In previous work, individual performance indicators are identified, in this work we identify a group of interrelated performance indicators. Finally, in this work a performance model is identified for each group of teams and a comparative analysis is carried out, identifying differences between the two, an aspect that also differs from the cited works.
Reviewer
The discussion section, the authors describe or report the results obtained, again similar to those reported by previous referenced studies (it is correct, they justify them), but they do not debate or discuss the reasons for those results or no indications are given of how they could be applied by coaches. Example paragraphs:243-253,254-267, 293-300…
Authors.
The information regarding the application of the results has been expanded. The explanation of some of the results is indicated, those results that are not justified are explained because a possible cause has not been found. Based on the comments of the other reviewer, we consider that the rest of the content in this section is adequate.
Reviewer
Review table 1: goals conceded.
Authors
Thank you very much for the appreciation. We have modified it.
Reviewer
Line 276-277: The authors could justify that 4 points is a “slightly greater weight”
Authors
The component "passes" is the one that shows the greatest difference in weight between the two groups, but despite this, this difference cannot be considered very important, 27% in best teams and 23% in bottom teams. We understand that we cannot talk about important differences, since it would not be the same as talking about a difference of 10-20 points. The differences are shown in the loadings of each component.
Reviewer
Line 286: References.
Authors
We do not understand what you mean by this comment. If what you want to indicate to us is that the section is not numbered, in that case, we indicate that this section is not numbered, following the instructions in the journal.
Reviewer
Based on the journal's indications, this section should provide readers with a brief summary of the main conclusions. Therefore, we consider that the information contained in this work is correct, trying to summarize the main findings of the work. Despite this we have made some changes to try to improve it.
Reviewer 2 Report
Please see file attached for my review of the manuscript.

Author Response
Dear reviewer, thank you very much for your comments and suggestions, we indicate the changes we have made based on the review made.
Reviewer
Lines 48-49: Good objective to classify teams by position at end of the season rather than match outcome.
Lines 69-74: Good justification using PCA
Authors
Thank you for your comments
Reviewer
Lines 85-86: Very good sample size
Authors
Thank you very much
Reviewer
Lines 105-107: Good method to ensure data reliability. Must have been quite time consuming! Although football matches are fun to watch of course
Authors
Thank you very much, we are sure that you know what you are talking about. Indeed, the recording of data is a very important part of the work, in terms of the time used to carry out this task.
Reviewer
Lines 119-128: Nice description of PCA.
Authors
Thank you
Reviewer
Equation (2) – the parameters (e.g. Yt, X1) should be defined clearly under the equation
Authors
This information was included
Yt explained variable “Effectiveness”
Xn Principals components of each group
Reviewer
Line 157: Why did you choose PC’s that only explain 70.1% of the variance? Typically 90-95% is used
Authors
The number m of principal components is obtained so that Pm is close to a value specified by the user, for example, between 70% and 80% is usual according to the consulted literature [see GORSUCH, RL (1983). Factor Analysis. Hillsdale, NJ: Lawrence Erlbraum Associates A tutorial on Principal Components Analysis, Lindsay I Smith February 2002.] We have decided to use 70% to be more restrictive and obtain the least number of components.
Reviewer
Lines 152:154 – Using the variables to predict outcome is a good way to validate they are meaningful indicators of performance. However, why did you create a performance variable effectiveness to validate? It is clear in your principal component weightings that goals scored/conceded etc. did not have a good loading compared to others features. I would recommend using the principal components in a supervised classification task, e.g. create feature vectors using your principal components and then having league table position finish as the target variable. Then randomly split the data and train classification models and evaluate on unseen data. This way you could see if the principal components are effectively classifying league position finish.
Authors
We decided to create the “effectiveness” variable, because we consider it to be the one that best defines performance in the offensive phase, in such a way that we use it as an explained variable to identify the most effective game model. Thank you for the design proposal that you propose, we consider it very interesting, and we will consider it for future work, however, it would be impossible to include it in the present work, since it would mean having to do a new work.
Reviewer
Figure 1- Screen plot best teams – On my version of the manuscript, the percentage loading for PC1 is cut out (27%). Can only see from 14.9% onwards, you may want to check and fix this.
Authors
Thank you very much for the comment, we have corrected it
Reviewer
Figures 2a – Figures2h – I assume the percentage contribution for each feature is calculated by summing the loadings for each feature-type and then dividing the individual feature by these sums. However the authors should state this in the manuscript.
Authors
Effectively, the percentage contribution for each characteristic is calculated by adding the charges for each type of characteristic and then dividing the individual characteristic by these sums. The contribution percentages of each characteristic to the different components can be observed in tables 3 and 5. We consider that this information is obvious and the results are provided by the statistical program itself, therefore, we believe that it is not necessary to include this information. Despite this, if you continue to consider that we should include it, we will include it.
Reviewer
Why was shots and shots on target including in the passes component? I would assume these belong in the shots component. Moreover, why is corners in the shots component, should this not be in passes.
Authors
The categories/variables of each component are provided by the PCA analysis, all we have done is subjectively assign a name to each component, taking into account the number of categories of the same or similar nature, and the weight of each category in each component.
Reviewer
It’s an interesting result that goals scored and goals conceded did not have a greater contribution to components. There must have been teams that were scoring a lot of goals but also conceding a lot? I performed a similar investigation on Australian Football League (AFL) data, albeit I ranked features with the target variable of match outcome and thus goals scored/conceded were very highly ranked.
So we ended up removing features directly related to scoring as it is common knowledge that a team which scores more and concedes less in the game would win. It seems that features directly related to scoring when using league position as the target variable do not have as great effect as it could be the case that teams are scoring a lot but also conceding a lot (as I said earlier).
Authors
Thank you very much for your comments, we will review your work and consider your suggestions in future work.
Reviewer
Table 5. In my version of the manuscript Table 5 and table 3 have a different format (table 5 does not have horizontal dividing lines). This should be kept consistent.
Authors
Thank you very much for the appreciation. We have modified it
Reviewer
Discussion
Good discussion of results, passing accuracy is obviously very important! But it is nice to have this quantified. Also good relation to previous scientific literature.
Authors
Thank you very much for your comments.
Reviewer
Conclusion
Lines 353-356 this seems like a very obvious conclusion (especially the number of goals part). But again, quantification of this is good. It would be good to add future research avenues to the conclusion, do you plan to look at the feature importance for individual teams at the top and bottom (e.g. Barcelona, Real Madrid, and Atletico Madrid at the top) and that way individual profiles for each team could be created. Then the bottom teams know what the golden standard is to aim for. Although it is well known that Atletico Madrid have a much more dogged/defensive style than Barcelona and Real Madrid. However I think this would be still an interesting analysis.
Authors
Indeed, the inclusion of the number of goals scored as an offensive performance factor is obvious, therefore, we have decided to eliminate it from the conclusions, since it does not provide relevant information.
Future lines of research are included in the discussion section, lines 344-349. We agree that it can be interesting to identify the individual profiles of successful teams, but our objective in this study was to carry out a nomothetic study to identify the playing patterns of soccer and not of a particular team. As you correctly point out, in soccer, success can be achieved with different play styles, and therefore our concern is to identify the most successful general models of the game, not those of a particular team. But we agree that it would be another interesting type of work. Thank you very much for your suggestion.
Reviewer
Summary
Overall, the presented manuscript is well presented and the methodology, statistical analysis is good. I would suggest assessing Figures 2a-2h as I believe some of the features do not belong in their respective components (shots in passing component). Also, due to the conclusion in line 353-356, I would consider removing features directly related to scoring (goals scored) from the feature set and seeing what the top ranking features for offensive success are then. As goals scored is quite obvious and coaches would know this already (doesn’t add much benefit to performance analysis). However removing these features may reveal important performance indicators that go unnoticed – which of course is the main purpose of this analysis. This paper is of great benefit to the performance community and after addressing my concerns, I believe is acceptable for publication. Also a minor review of the English is recommended.
Authors
Thank you for your suggestions. As we indicated previously, the variables of each component are determined by the PCA and cannot be eliminated, although some do not have a close relationship with the rest of the variables of the component. Also indicate that the exclusion of the goals would mean having to carry out all the analyzes again and, consequently, carry out all the method, results and discussion part again. We believe that at this stage of the work it would be something unfeasible. However, we will consider your suggestions for future work.
Regarding the English revision, the work has been translated by a native speaker and has already been revised a couple of times. It would be of great help if you could specify some specific errors.